# A retrospective analysis of cases of Spontaneous Bacterial Peritonitis in cirrhosis patients

**Phillip Pasquale Santoiemma** [1,2], **Omar Dakwar** [2], **Michael Peter Angarone** [1,2] *

**1** Department of Internal Medicine, Northwestern University Feinberg School of Medicine, Chicago, IL, United States of America, **2** Division of Infectious Diseases, Northwestern University Feinberg School of Medicine, Chicago, IL, United States of America

\* m-angarone@northwestern.edu

**Data Availability Statement:** All relevant data are within the manuscript and its Supporting Information files.

**Funding:** The funders had no role in study design, data collection and analysis, decision to publish, or preparation of the manuscript. Award Received by:

## Abstract

### Background & aims

Spontaneous Bacterial Peritonitis (SBP) is an infection in patients with cirrhosis and carries significant mortality. The management of SBP is evolving with the rise of multidrug resistant organisms. Our aim was to perform a retrospective analysis to determine if identification of bacteria in culture could aid in prognosis and provide information regarding optimal treatment.

### Methods

We analyzed our 10-year experience of SBP in a single academic center (Northwestern Memorial Hospital). We obtained information regarding SBP prophylaxis, culture data and resistance patterns of bacteria, choice/duration of inpatient antibiotics, and key laboratory measurements and determined outcomes including mortality, hospital duration, and ICU stay.

### Results

Patients with SBP had a 17.8% mortality and had culture positive SBP 34.4% of the time. Antimicrobial resistance was seen in 21.3% of cases and trended towards worsening mortality, with worsened mortality associated with first line use of piperacillin-tazobactam (p = 0.0001). Patients on SBP prophylaxis who developed SBP had improved mortality (p<0.0001) unless there was a positive culture, in which case patients had worsened mortality (p = 0.019). Patient with a higher PMN counts after repeat paracentesis had higher mortality (p = 0.02).

### Conclusions

Our results show that SBP continues to be a morbid and deadly condition and identification of an organism is key in treatment. The standard initial antibiotic for SBP may need to be modified to reflect emerging resistant pathogens and gram-positive organisms. Further,

PS and MA The work was supported by, in part, by the National Institutes of Health's National Center for Advancing Translational Sciences (Grant Number UL1TR001422) and the Northwestern Medicine Enterprise Data Warehouse (NMEDW).

**Competing interests:** The authors have declared that no competing interests exist.

antibiotic prophylaxis should be utilized only in select cases to prevent development of resistance.

## Introduction

Spontaneous Bacterial Peritonitis (SBP) is an infection in the peritoneum that develops in patients with ascites related to cirrhosis. SBP remains an important complication in persons with cirrhosis and is a significant cause of mortality, with rates between 20–40% in decompensated cirrhosis [1] and as high as 75% in some cohorts [2–4]. The mechanism of SBP is poorly understood and is thought to be secondary to bacterial translocation from the gut, reduced gut motility leading to bacterial overgrowth, and/or altered intestinal defenses and immune responses [5, 6]. The most common organisms causing SBP arise from the gastrointestinal tract and include *Escherichia coli*, *Klebsiella pneumoniae*, *Streptococcus* species (spp.) and *Enterococcus* spp. [7].

The incidence of SBP has been estimated to be between 10 and 30% in chronic liver disease patients [8], with up to 3% occurring in the outpatient setting [6, 9]. Laboratory diagnosis of SBP is established by identifying an elevated ascitic fluid polymorphonuclear cell (PMN) count greater than or equal to 250 cells/ul and a positive culture of the ascitic fluid. Without a positive culture, the diagnosis is referred to as Culture Negative Neutrocytic Ascites (CNNA) and SBP with a positive culture and a PMN threshold less than 250 is referred to as Non-Neutrocytic Bacteria-Ascites (NNBA). SBP, CNNA and NNBA are often used interchangeably. A recent retrospective review demonstrated no difference in mortality in patients with SBP vs. NNBA [10]. With proper technique, up to 60% of peritoneal fluid samples will be positive for a bacterial pathogen [11–14].

SBP prevention is recommended for higher risk cirrhotic patients, such as those with low ascitic fluid protein, risk factors for hepatorenal syndrome, and a prior history of SBP. Oral fluoroquinolones are currently the prophylactic antibacterial of choice [15, 16] Empiric therapy for SBP typically relies on the use of third-generation cephalosporins and when possible therapy should be targeted for the cultured organism [17, 18]. Treatment is often modified or continued based on a follow-up paracentesis done on day 3 of treatment [19], although there is limited data on repeating paracentesis and this strategy may not improve mortality [20]. Resistance to standard first line agents and prophylaxis is high, ranging from 33–49% [7, 11, 21] of cases, and continues to increase [22–25]. Infection with a resistant bacteria portends a poor prognosis [26].

Our project is a single center review on the causative agents of SBP in patients with cirrhosis. We reviewed data among patients with cirrhosis and SBP from a single academic center and identify a sub-set of risk factors that could potentially change standard treatment and management of SBP. Our aim was to determine the incidence, mortality and morbidity of SBP, the percentage of patients with positive bacterial cultures, the types of bacteria grown from those cultures and the morbidity/mortality of each individual bacteria, the change in Day 1 and Day 3 paracentesis results, and the resistance rates to standard treatment and prophylaxis. We hypothesized that identification of bacteria in culture could aid in prognosis and provide targeted treatment given resistance patterns to first line agents.

## Patients and methods

We performed a chart review using the Northwestern Enterprise Data Warehouse(EDW) with the approval of the Northwestern Biomedical IRB (STU00204726). Consent: Consent order was waived by approved IRB. Rationale: The study was retrospective and involves collection of

data elements, which have already been obtained as part of clinical care. We queried the database to report all patients who underwent paracenteses at Northwestern Memorial Hospital in the inpatient and outpatient settings from January 1st, 2006 to December 31st, 2016.

Inclusion criteria for review were patients with a diagnosis of SBP, cirrhosis (identified by ICD-9, ICD-10 and CPT codes) and adults between the ages of 18 and 88 years old. We excluded patients <18 years old and >88 years old, individuals undergoing post-surgical paracentesis, individuals with HIV infection, patient who received a stem cell transplant, and those without cirrhosis at the time of the procedure. We reviewed the Epic Electronic Medical Record system to verify information from EDW queries in all cases (Fig 1).

Demographic information including age, gender, and ethnicity, date of paracentesis, and etiology of cirrhosis diagnosis was extracted from each chart. Information regarding the details of the diagnosis of SBP from the paracentesis were obtained via chart review and included PMN count, culture data and species/resistance patterns. Data related to agent used for SBP prophylaxis, the choice/duration of inpatient antibiotics for treatment of SBP, whether changes were made in antibiotic choice, and key laboratory measurements at the time of diagnosis were analyzed from the subjects' chart. Outcomes of patient stay were determined including hospital duration, all-cause mortality, ICU stay, in- hospital and recurrence of SBP.

Descriptive statistics were produced using frequencies to describe the SBP cohort. To assess the differences in demographics and other hospitalization factors between those with positive SBP cultures and negative SBP cultures, Student's t-tests and Chi-square tests were performed. Other univariate testing was done to assess types of bacteria in positive cultures, drug resistance, and prophylactic antibiotics. Statistical significance was set at $p<0.05$. Statistical analyses were performed using Statistical Analysis System (SAS) version 9.4 (SAS Institute Inc., Cary, North Carolina, USA) and Microsoft Excel 2019.

## Results

In total, 2159 patients were identified as having cirrhosis and undergoing a paracentesis over the 10-year study period. Of these subjects, 314 patients were diagnosed and treated for SBP at Northwestern Memorial Hospital for a rate of 14.5% (Table 1).

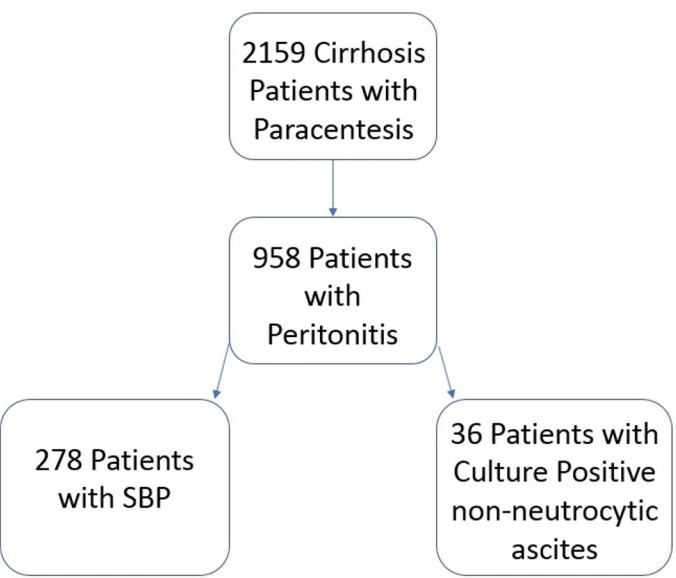

**Fig 1. Stepwise process of data mining starting with EDW patient data.**

**Table 1. Descriptive statistics of patients in the SBP cohort.**

|  | Culture Positive | Culture Negative |  |
|---|---|---|---|
|  | n = 108 | n = 206 | P |
| *Average Age at SBP diagnosis (Years, Range 20–87)* | 57.8 | 56.1 | 0.177 |
| *Gender* | 62.4% Male, 37.6% Female | 63.4% Male, 36.6% Female | 0.857 |
| *Etiology of Cirrhosis* | 75.2% Non-Alcoholic Cirrhosis | 67.3% Non-Alcoholic Cirrhosis | 0.169 |
| *Race* | 65% Caucasian, 10% Hispanic, 8% African-Amer, 3% Asian, 14% Other | 54% Caucasian, 18% Hispanic, 7% African-Amer, 2% Asian, 19% Other | 0.203 |
| *Average Hospital Length of Stay (Days)* | 13.1 | 11 | 0.107 |
| *Mortality during Hospitalization* | 25.7% | 13.7% | **0.008**\* |
| *Transfer to ICU during Hospitalization* | 52.3% | 36.1% | **0.007**\* |
| *Average MELD-Na* | 28.5 | 25.9 | **0.009**\* |
| *Percentage on prophylactic antibiotics* | 27.5% | 19.0% | 0.083 |

## Bacterial culture data

Among all episodes of SBP, 34.4% were culture positive and yielded 27 unique organisms, with 50.4% being gram negative organisms (Table 2). There were no significant differences in culture positive and culture negative patients in terms of age, gender, etiology of cirrhosis, race, length of stay and percentage of patients on prophylactic antibiotics; there were significant differences in mortality (25.7% vs. 13.7%, p = 0.008), ICU Transfer rates (52.3% vs. 36.1%, p = 0.007) and MELD-Na score (28.5 vs. 25.9, p = 0.009) (Table 1).

**Table 2. Details on positive culture including bacteria, mortality and susceptibility to ceftriaxone or piperacillin-tazobactam.**

| Bacteria | Frequency of Positive Culture | Mortality | ICU Stay | MELD-Na | Susceptibility to Ceftriaxone | Susceptibility to Piperacillin-Tazobactam |
|---|---|---|---|---|---|---|
| **Gram-negative bacteria** |  |  |  |  |  |  |
| *Escherichia coli* | 28 | 36% | 50% | 30 | 25/28 | 26/27 |
| *Klebsiella pneumonia* | 18 | 17% | 56% | 28 | 15/18 | 11/12 |
| *Klebsiella oxytoca* | 4 | 25% | 25% | 25 | 4/4 | 3/3 |
| *Citrobacter spp.* | 4 | 0% | 75% | 26 | 4/4 | ND |
| *Corynebacterium spp.* | 3 | 33% | 67% | 30 | ND | ND |
| *Pseudomonas aeruginosa* | 2 | 0% | 100% | 43 | 0/2 | 1/1 |
| *Enterobacter cloacae* | 2 | 0% | 100% | 20 | 1/2 | 1/2 |
| *Serratia marcescens* | 1 | 100% | 100% | 20 | 1/1 | ND |
| *Acinetobacter spp.* | 1 | 0% | 0% | 23 | 1/1 | ND |
| *Proteus mirabilis* | 1 | 0% | 0% | 17 | 1/1 | 1/1 |
| **Gram-positive bacteria** |  |  |  |  |  |  |
| *Streptococcus spp.* | 21 | 10% | 14% | 27 | 14/16 | ND |
| *Enterococcus spp.* | 15 | 33% | 73% | 27 | 0/15 | 7/7 |
| *Other Staph spp.* | 9 | 22% | 56% | 28 | ND | 2/2 |
| *Staphylococcus aureus* | 3 | 0% | 67% | 30 | 0/1 | 2/3 |
| **Total/Average** | 112 | 22% | 56% | 28.2 | 66/93 (71%) | 52/59 (88%) |

Note not all isolates were tested against ceftriaxone and/or piperacillin-tazobactam. Of note, susceptibility of *Enterococcus* spp. to piperacillin-tazobactam were extrapolated from ampicillin susceptibilities.

**Table 3. Mortality and ICU stay comparisons for gram type among the most common organisms (*E. coli*, *K. pneumoniae* and *Enterococcus spp*).**

| Death | Gram | | Total |
|---|---|---|---|
| | Positive | Negative | |
| Yes | 6 | 13 | 19 |
| No | 18 | 33 | 51 |
| Total | 24 | 46 | 70 |

Chi-Square = 0.08 p-value = 0.77

Note that there was no statistical difference in mortality for Gram Type but the most three most common bacteria trended towards worsened mortality (p = 0.08) but were significantly more likely to lead to ICU transfer (data not shown) (p = 0.01).

The four most common organisms include *Escherichia coli* n = 28, *Klebsiella pneumoniae* n = 18, *Enterococcus faecium* or *faecalis*. n = 15, and viridans group *Streptococcus* n = 9. Of note, one patient grew *Clostridium perfringens* and two grew *Candida spp*. which were not included in the overall analysis. These organisms were included in analysis to determine mortality and morbidity (Table 3). There was no statistically significant difference in mortality between Gram Positives vs. Gram Negatives or between specific organism, although common organisms trended towards significantly worse mortality (p = 0.08, Table 3). The *three* most common organisms were associated with increased ICU transfer (p = 0.01) and increased length of stay (p<0.001). Infection with a gram-negative organism was associated with an increased length of stay (p = 0.002). The MELD-Na score was not statistically different between Gram Positive vs. Gram Negative organisms or between specific organisms.

## Antimicrobial resistance

Among the bacteria isolated, 27 (21.3%) were resistant to at least one first line antibacterial agent for the treatment of SBP. Of the 27 resistant bacteria isolated 15 were identified as *Enterococcus* spp. (bacteria inherently resistant to ceftriaxone), 9 were gram negative bacteria and two were *Streptococcus* species. Of those with a gram negative isolate resistant to ceftriaxone, 3 also had resistance to piperacillin-tazobactam. Patients with drug resistant bacteria had a mortality rate of 25.9% compared to 22.3% mortality rate for any positive culture (Table 4). Patient with positive cultures and drug resistant bacteria were more likely to be transferred to the ICU (p = 0.026) compared to the main cohort.

## First-line antibiotic treatment

Ceftriaxone and piperacillin-tazobactam were the most frequently used empiric antibiotics. Empiric antimicrobials were changed in 47.8% of cases and broadening of antibiotics was not

**Table 4. Comparison of all patient with SBP, patients with positive cultures and patient with resistant organisms.**

| Patient Characteristics | All SBP Patients | Culture Positive SBP | Cultures with Drug Resistant Bacteria | P |
|---|---|---|---|---|
| *Number of Patients* | 314 | 108 | 27 | |
| *Age (years)* | 56.7 | 57.8 | 58.6 | 0.33 |
| *MELD-Na* | 26.7 | 28.5 | 28.5 | 0.393 |
| *Mortality* | 17.8% | 25.7% | 25.9% | 0.298 |
| *ICU Transfer* | 41.4% | 52.3% | 66.7% | 0.028 |

Of note, drug resistant bacteria implies drug resistance to either ceftriaxone or piperacillin-tazobactam.

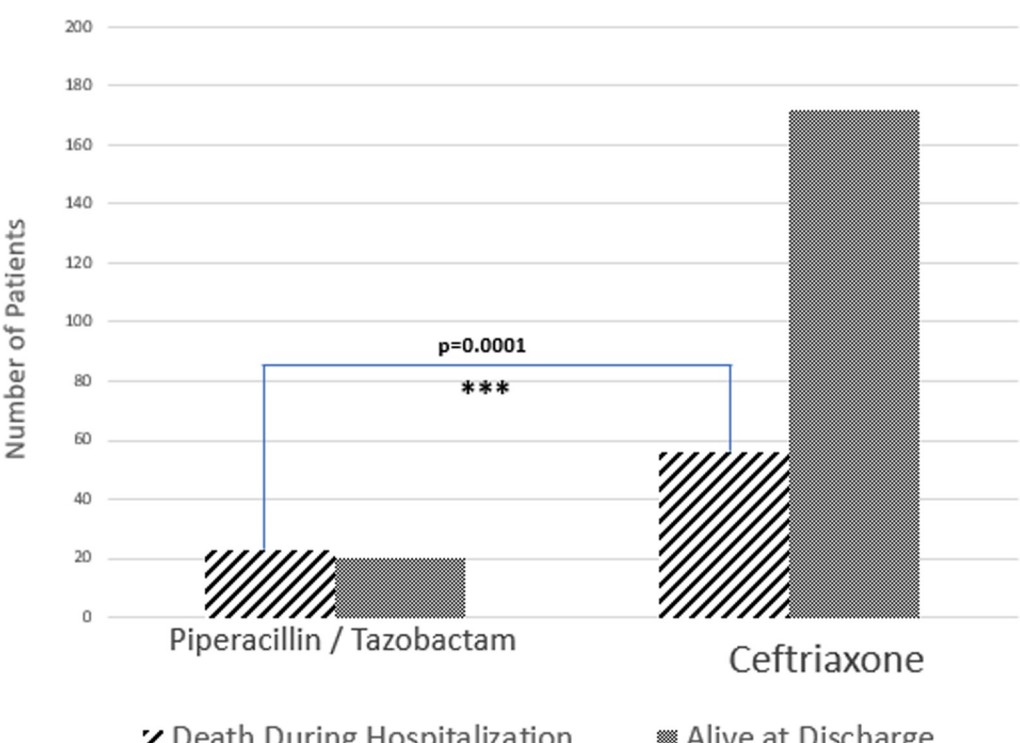

**Fig 2. First line antibiotic choice.** Using Piperacillin/Tazobactam as the first line agent leads to statistically worsened mortality (p = 0.0001).

associated with a change in mortality (p = 0.16). Use of piperacillin-tazobactam was associated with higher mortality (29.1% compared to 10.4%, p = 0.0001, Fig 2), transfer to an ICU (62.0% compared to 29.7%, p<0.001) and longer length of stay (p<0.001) without any statistically significant difference in MELD-Na (p = 0.056).

## SBP prophylaxis

In our cohort of patients, 22% of patients (69 total) were on SBP prophylaxis (most commonly ciprofloxacin) prior to diagnosis of SBP. Among these 69 patients, 30 were culture positive, with 16 out of 30 (53%) being resistant to the prophylaxis used (Table 5).

Overall, patients who developed SBP while on prophylaxis had improvement in overall mortality (p<0.0001) compared with patients not on prophylaxis. However, the development of culture positive peritonitis while on prophylaxis was associated with higher mortality when compared to all patients not on prophylaxis (p = 0.019).

**Table 5. Analysis of patients on prophylactic antibiotics.**

|  | Number of Patients | MELD-Na | Age (years) | Mortality | P | ICU Transfer | p |
|---|---|---|---|---|---|---|---|
| **Patients not on Prophylaxis** | 245 | 26.6 | 57.6 | 18.3% | n/a | 40.8% | n/a |
| **Patients on Prophylaxis** | 69 | 27.2 | 53.6 | 15.9% | < **.00001**\* | 43.5% | 0.692 |
| *Culture Positive* | 30 | 26.6 | 53.4 | 36.7% | **0.019**\* | 50.0% | 0.336 |
| *Resistant Bacteria to Prophylaxis* | 11 | 29.1 | 59.8 | 18.2% | 0.988 | 45.5% | 0.76 |
| *Sensitive Bacteria to Prophylaxis* | 8 | 28.3 | 46.1 | 25.0% | 0.635 | 62.5% | 0.221 |

Mortality and ICU transfer rate were compared between patients not on prophylaxis and subsets of groups of patients on prophylaxis.

### Repeat paracentesis data

Most patients (77.1%) underwent a repeat paracentesis at Day 3 to determine a repeat PMN count. A decrease in the PMN count was seen in most patients, however 42 patients (17.4%) had a higher Day 3 PMN count. Therapy was changed in 72% of patients regardless of the repeat peritoneal fluid PMN count. A majority of patients started on ceftriaxone (84%) had their antibacterial changed, compared to 53.6% of patients empirically started on piperacillin-tazobactam (p = 0.045). Patients with a higher day 3 PMN count in the peritoneal fluid had higher mortality rates compared to those that had a decrease in the PMN count (17.3% compared to 13.7%, p = 0.02). There was no statistically significant difference in ICU transfer (p = 0.38), length of stay (p = 0.39), and MELD-Na (p = 0.75) in patients with higher compared to lower day 3 paracentesis counts. Patients who did not undergo a repeat paracentesis had a mortality of 27.8%, significantly higher than patients who had repeat paracenteses.

## Discussion

Our results indicate SBP continues to be a very challenging infection to treat and carries with it a high mortality rate (17.8% in our review). Culture positive peritonitis was seen in 34.4% of patients, with an equal distribution between Gram Positive and Gram-Negative organisms. Culture positive peritonitis was associated with higher MELD-Na score, rate of ICU transfer and mortality.

Antimicrobial resistance was seen in 27 cases of peritonitis, to either ceftriaxone or pipera-cillin-tazobactam. Antimicrobial resistance was associated with ICU transfer and a trend towards worsening mortality. First line use of piperacillin-tazobactam was associated with significantly worsened mortality. Worse mortality was also seen in patients who had a higher PMN count on repeat paracentesis or whom never had a repeat paracentesis.

A majority of episodes of SBP in our study were culture negative. The most common pathogens identified in our study were *Escherichia coli* and *Klebsiella pneumonia*, in agreement with the bacteriology of SBP seen in the literature [7, 27], although recent studies demonstrate a trend toward more Gram Positive organisms [28, 29]. In our study, patients who had positive bacterial cultures had significantly worse morbidity and mortality. A potential explanation for this finding is that patients who have positive cultures are more ill due to worsened gut translocation, poorer immune defenses, or lack of antibiotic prophylaxis. Further research into improved diagnostics for SBP, namely use of multiplex polymerase chain reaction (PCR) bacterial panels, should be undertaken to help identify pathogens and potential resistance patterns quicker which should lead to earlier targeted therapy.

Patients receiving prophylaxis for SBP had overall improved mortality compared to patients not on prophylaxis. However, those with a positive bacterial culture on prophylaxis had worsened mortality. A likely explanation for this finding is selecting for more resistant bacteria, as multi-drug resistant organisms are on the rise in SBP [27, 30]. Rostkowska et al. presents data that long term use of fluoroquinolones as prophylaxis may increase the risk of ESBL producing *Enterobacter* spp. by 4 fold [30]. Further research will be needed to determine if prophylaxis leads to enough resistance to warrant cessation of this practice.

Of the patients with a positive bacterial culture, 21% had antimicrobial resistance to first line SBP treatment, similar to what has been found in the literature [11, 20, 28, 31, 32]; these patients tended to be sicker and had a trend towards worsened mortality. Lutz et al. describes a series of 86 patients with SBP in which resistance to initial antibiotic treatment worsened 30 day mortality from 18% to 68% [32]. Oliveira et al. performed a retrospective study of 113 patients with SBP which found 46.9% with multidrug resistant bacteria, including 39% resistance to third generation cephalosporins [31]. Our data as well as the literature demonstrate

the value of culture data in management of SBP by allowing for targeted therapy based on bacterial sensitivities and potentially affecting outcomes. Clinicians managing patients who have cultures that grow resistant bacteria should modify therapy and be mindful of the worsened prognosis seen in these individuals.

Performing a repeat paracentesis is still a valuable tool in the management of SBP. Patients found to have worsening PMN values on repeat paracentesis, as well as those that did not have a repeat paracentesis, have a higher mortality than patients with improving PMN numbers. The latter of these findings is unclear but may indicate patients who may have been too unstable to get a repeat procedure or may have already been clinically worsening. Repeat paracentesis often led to significant management changes, either based off of culture data or PMN values. Thus, we recommend continuing to perform a repeat paracentesis on day 3 of patient care in agreement with recent recommendations [20].

In our cohort broader initial therapy was not better therapy. Patients who received piperacillin-tazobactam as first line therapy had worsened mortality compared with patients receiving ceftriaxone. One obvious reason for this would be that the patients receiving piperacillin-tazobactam were sicker, thus warranting broader initial treatment. However, there was no difference in patients receiving either antibiotic in terms of MELD-Na, meaning that the overall illness level of these patients was relatively similar. Further, all of these patients were suitably stable for management on a general medicine or hepatology floor on admission, which suggests a similar level of illness throughout the cohort, although we did not chart blood pressure or lactate in our analysis.

Our data bring into question the use of standard first line regimens in SBP and whether the first line antibiotic should account for multi-drug resistant organisms not covered by ceftriaxone or piperacillin-tazobactam. Although these agents are still effective for the majority of patients, two factors drive the ineffectiveness of these medications: 1). The emergence of gram-positive bacteria not covered by these agents and 2). multi-drug resistant pathogens that are on the rise. Our results support the growing literature in SBP [32, 33] that indicate that the empiric antibiotic therapy for SBP likely needs to be changed [27, 31, 34]. Further research needs to be conducted in order to determine if extended spectrum antibiotics, e.g. carabapenems, or the addition of resistant gram-positive therapy, e.g. daptomycin, should be used as first line therapy.

Our research was limited to a single center study and may not necessarily be generalizable. Further, the study was retrospective and occurred over a period of 10 years, of which there have been many changes in SBP and cirrhosis management (more prevalent antibiotic prophylaxis, improvements in cirrhosis treatment, recent emergence of more resistant pathogens, etc.) which may have affected the result of the work.

## Conclusions

Our study demonstrates the importance of choice of initial therapy, identification of a causative organism and the impact of prophylaxis on the outcomes of SBP. Further, our results demonstrate that while prophylaxis is beneficial for patients with cirrhosis who are at higher risk for SBP, it may impact morbidity if patients develop SBP.

Antimicrobial resistant pathogens, as demonstrates by our results as well as others, are playing a more important role in SBP and outcomes from SBP. Choice of antibiotics should consider local antibiotic resistance patterns to ceftriaxone or piperacillin-tazobactam. Perhaps the use of more modern pathogen identification (rapid, multiplex polymerase chain reaction searching for the most common SBP bacteria for instance) would help to aid in targeted therapy for patients.

## Supporting information

**S1 Dataset.**
(XLSX)

## Acknowledgments

We would like to acknowledge the statisticians at the Northwestern Electronic Data Warehouse for assisting in data collection.

## Author Contributions

**Conceptualization:** Phillip Pasquale Santoiemma, Michael Peter Angarone.

**Data curation:** Phillip Pasquale Santoiemma, Omar Dakwar, Michael Peter Angarone.

**Formal analysis:** Phillip Pasquale Santoiemma, Omar Dakwar, Michael Peter Angarone.

**Funding acquisition:** Phillip Pasquale Santoiemma.

**Investigation:** Phillip Pasquale Santoiemma.

**Methodology:** Phillip Pasquale Santoiemma, Michael Peter Angarone.

**Project administration:** Phillip Pasquale Santoiemma, Michael Peter Angarone.

**Software:** Phillip Pasquale Santoiemma, Omar Dakwar.

**Supervision:** Michael Peter Angarone.

**Validation:** Phillip Pasquale Santoiemma.

**Visualization:** Phillip Pasquale Santoiemma.

**Writing – original draft:** Phillip Pasquale Santoiemma.

**Writing – review & editing:** Phillip Pasquale Santoiemma, Omar Dakwar, Michael Peter Angarone.

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
