## [Decision Letter · Decision Letter 0]

21 Jul 2020

PONE-D-20-15175

A Retrospective Analysis of cases of Spontaneous Bacterial Peritonitis in Cirrhosis patients

PLOS ONE

Dear Dr. Angarone,

Thank you for submitting your manuscript to PLOS ONE. After careful consideration, we feel that it has merit but does not fully meet PLOS ONE’s publication criteria as it currently stands. Therefore, we invite you to submit a revised version of the manuscript that addresses the points raised during the review process.

A number of gaps in data presentation and interpretation have been pointed out. These need to be addressed. 

We look forward to receiving your revised manuscript.

Kind regards,

Iddya Karunasagar

Academic Editor

PLOS ONE

Journal Requirements:

Additional Editor Comments (if provided):

Two reviewers have commented on the manuscript and a number of shortcomings have been pointed out. Please address all comments point by point by point.

Reviewers' comments:

Reviewer's Responses to Questions

**Comments to the Author**

1. Is the manuscript technically sound, and do the data support the conclusions?

Reviewer #1: Partly

Reviewer #2: Yes

2. Has the statistical analysis been performed appropriately and rigorously? 

Reviewer #1: N/A

Reviewer #2: Yes

3. Have the authors made all data underlying the findings in their manuscript fully available?

Reviewer #1: No

Reviewer #2: Yes

4. Is the manuscript presented in an intelligible fashion and written in standard English?

Reviewer #1: Yes

Reviewer #2: Yes

5. Review Comments to the Author

Reviewer #1: 1. The details of all laboratory investigations done on these patients should have been included. Was ascitic total protein concentration measured ?

2. The complete antibiogram of the isolates has not been provided. Results of susceptibility to two antibiotics (Ceftriaxone and piperacillin-tazobactam) alone are given. Majority of the patients were treated with these two antibiotics. What about the remaining patients ? What antibiotics were they treated with and what was the susceptibility profile ?

3. What was the daily dose of Ceftriaxone ?

4. How many patients had a PMN count of more than 500 ?

5. What could be the reason for increased mortality in patients who were treated with piperacillin-tazobactam.

Why did piperacillin-tazobactam continue to be used as empiric therapy in spite of the fact that it increased mortality rates ?

Is piperacillin-tazobactam recommended according to the practice guidelines ?

6. Did practice guidelines for SBP change during the course of 10 years of study duration ?

7. The hypothesis that identification of bacteria in culture could aid in prognosis and provide targeted treatment is already known and proven.

Reviewer #2: Dear Authors

The paper describes a retrospective analysis for 10 years of spontaneous bacterial peritonitis in cirrhosis in a single center.

The paper shows data on a sizable population.However, the following needs to be addressed

1. The authors mention mortality but do not mention time frame. Is this 30 day mortality data that is limited to the hospitalized patients?

2. The authors mention the changes in the guidelines during the 10 year period. But what changes were made in the department policy in line with the newer guidelines?

3. What is the inference of this study? Do you recommend that the policy needs to be institutional or regional as antibiograms and organism profiles vary between different hospitals

4. The recommendation of use of cefotaxime/ ceftriaxone needs addressal here with such a large number of isolates of Emterococcus seen

5. What were the other parameters looked at? Albumin?

6. Culture technique for these cases also needs to be mentioned.

6. PLOS authors have the option to publish the peer review history of their article (what does this mean?). If published, this will include your full peer review and any attached files.

Reviewer #1: **Yes: **Thangam Menon

Reviewer #2: No

---

## [Author Response · Author response to Decision Letter 0]

25 Aug 2020

Comments

We note that you have included the phrase “data not shown” in your manuscript. Unfortunately, this does not meet our data sharing requirements. PLOS does not permit references to inaccessible data. We require that authors provide all relevant data within the paper, Supporting Information files, or in an acceptable, public repository. Please add a citation to support this phrase or upload the data that corresponds with these findings to a stable repository (such as Figshare or Dryad) and provide and URLs, DOIs, or accession numbers that may be used to access these data. Or, if the data are not a core part of the research being presented in your study, we ask that you remove the phrase that refers to these data.

Thank you for your response. We have removed the phrase that refers to these data. 

We have included all of our data in an excel file which we can upload. 

Reviewer #1: 

1. The details of all laboratory investigations done on these patients should have been included. Was ascitic total protein concentration measured?

Thank you for your comment. Yes, in addition to PMN count, culture data and species/resistance patterns, we obtained ascites total protein and albumin on almost all patients. Further, we obtained a number of basic lab values including blood counts, creatinine, liver function tests, etc. that were used to determine MELD score. We have updated our manuscript to include the fact that we determined ascites protein and albumin levels, and also included the data set where this information was obtained:

“Information regarding the details of the diagnosis of SBP from the paracentesis were obtained via chart review and included PMN count, ascites protein and albumin levels, culture data and species/resistance patterns.”

2. The complete antibiogram of the isolates has not been provided. Results of susceptibility to two antibiotics (Ceftriaxone and piperacillin-tazobactam) alone are given. Majority of the patients were treated with these two antibiotics. What about the remaining patients ? What antibiotics were they treated with and what was the susceptibility profile ?

This is a great point and we appreciate your response. The focus on our article was on the two primary drugs used to treat SBP, Ceftriaxone and Piperacillin-tazobactam. Most patients, unfortunately, did not have culture positive results. Most patients, due to the nature of the way the data was obtained as well as the fact not all patients had full susceptibility testing, do not have a full report on susceptibility on every bacteria. 

In our review, we did not focus on the patients who received alternative to standard of care antibiotics; these include patients that had allergies to first line agents, prior resistant pathogens, or when clinicians decided to trial other antimicrobials. We do not report when other antibiotics were used because there were very few cases, please see attached dataset for further review. . 

3. What was the daily dose of Ceftriaxone ?

At our institution, usually, standard dosing is 2 grams IV daily. Some providers choose 1 gram IV daily. 

4. How many patients had a PMN count of more than 500 ?

I think this is a very interesting comment. Per the definition of SBP, we use a cutoff of 250 to make the diagnosis. In reviewing the data, it appears most patients (around 85%) of patients with SBP had a PMN count of more than 500. We decided to follow the guideline definition of SBP for our analysis. 

5. What could be the reason for increased mortality in patients who were treated with piperacillin-tazobactam. Why did piperacillin-tazobactam continue to be used as empiric therapy in spite of the fact that it increased mortality rates ? Is piperacillin-tazobactam recommended according to the practice guidelines ?

Thank you for your questions. Our reasons for increased mortality are discussed in the discussion section; but mainly we are unclear as to the exact reason (see expanded response below).

 To answer your second question, this was a retrospective study and so it was unknown that piperacillin-tazobactam led to higher mortality. 

See next question for answer to your third part question.

“Unfortunately, we do not have a logical explanation for why a more broad initial regimen was inferior to a narrower antibiotic.”

6. Did practice guidelines for SBP change during the course of 10 years of study duration ?

Thank you for your question, as this is an excellent question that we struggled with. 

The practice guidelines have not drastically changed over the last 10 years on SBP. There has been an increase in the use of prophylactic antibiotics, but in general, first line therapy remains 3rd generation cephalosporins. With emerging data, including that from our study, we argue that empiric therapy likely needs to be changed given more resistant pathogens and likely clinicians will need to start broader in patients who are more likely to have resistant pathogens. We discuss this in the second to last paragraph of our discussion. 

We have added a few sentences in the discussion on this question with citations.

“Unfortunately, practice guidelines for empiric coverage for SBP have not drastically changed over the last 10 years. (along with the most recent guideline citation). “

7. The hypothesis that identification of bacteria in culture could aid in prognosis and provide targeted treatment is already known and proven.

It is true that having advanced diagnostics in terms of culture data is valuable; what was not known was how having positive cultures could change outcomes (better or worse, would this lead to changes in antibiotics, etc.) which is what we addressed in our paper. 

Reviewer #2: 

1. The authors mention mortality but do not mention time frame. Is this 30 day mortality data that is limited to the hospitalized patients?

Sorry this was not clear. We studied mortality during hospitalization. All of the patients were hospitalized so this is only if they had mortality during their hospital stay; we have clarified this in the text

2. The authors mention the changes in the guidelines during the 10 year period. But what changes were made in the department policy in line with the newer guidelines?

This is a great point. The practice guidelines have not drastically changed over the last 10 years on SBP. There has been an increase in the use of prophylactic antibiotics, but in general, first line therapy remains 3rd generation cephalosporins. With emerging data, including that from our study, we argue that empiric therapy likely needs to be changed given more resistant pathogens and likely clinicians will need to start broader in patients who are more likely to have resistant pathogens. We discuss this in the second to last paragraph of our discussion. 

We have made changes in our department as a result of our data. 

We have added a few sentences in the discussion on this question along with guideline citation. 

“Unfortunately, practice guidelines for empiric coverage for SBP have not drastically changed over the last 10 years. (along with the most recent guideline citation). “

3. What is the inference of this study? Do you recommend that the policy needs to be institutional or regional as antibiograms and organism profiles vary between different hospitals

That is a great question and point. Yes, especially for hospitalized patients, we believe that policy needs to be institutional (especially given the wide difference in antibiograms from all around the world), but need to encompass more broad treatment at the onset. We have added a sentence in the discussion to that very point. 

“We recommend that a policy be established for treatment of SBP on an institutional basis, especially for hospitalized patients, based on the hospitals antibiogram to encompass emerging resistance patterns. Perhaps, if patients are on prophylaxis, are clinically unstable, or have increased risk for resistant pathogens (i.e. frequent antibiotic courses, previously positive cultures with resistant bacteria), that they are started on a carbapenem as first line therapy. “

4. The recommendation of use of cefotaxime/ ceftriaxone needs addressal here with such a large number of isolates of Enterococcus seen.

This is an excellent point. We agree that treatment of ceftriaxone is likely too narrow for SBP given the fact that enterococcus is a somewhat common pathogen in our cohort. Again, practices likely need to change as addressed in your third point. 

5. What were the other parameters looked at? Albumin?

Yes, in addition to PMN count, culture data and species/resistance patterns, we obtained ascites total protein and albumin on almost all patients. Further, we obtained a number of basic lab values including blood counts, creatinine, liver function tests, etc. that were used to determine MELD score. The details on all of these lab values are in the supplemental data file. We have updated our manuscript to include the fact that we determined ascites protein and albumin levels. 

“Information regarding the details of the diagnosis of SBP from the paracentesis were obtained via chart review and included PMN count, ascites protein and albumin levels, culture data and species/resistance patterns.”

6. Culture technique for these cases also needs to be mentioned.

I agree, this is a great point. Certainly, we can add this statement to our methods section. 

“Patients who underwent paracentesis undergo sterile bedside procedure technique; aerobic and anaerobic culture bottles are inoculated at bedside and sent to the microbiology lab where the fluid is plated on culture media plates. “

---

## [Decision Letter · Decision Letter 1]

8 Sep 2020

A Retrospective Analysis of cases of Spontaneous Bacterial Peritonitis in Cirrhosis patients

PONE-D-20-15175R1

Dear Dr. Angarone,

We’re pleased to inform you that your manuscript has been judged scientifically suitable for publication and will be formally accepted for publication once it meets all outstanding technical requirements.

Kind regards,

Iddya Karunasagar

Academic Editor

PLOS ONE

Additional Editor Comments (optional):

All comments addressed

Reviewers' comments:

Reviewer's Responses to Questions

**Comments to the Author**

1. If the authors have adequately addressed your comments raised in a previous round of review and you feel that this manuscript is now acceptable for publication, you may indicate that here to bypass the “Comments to the Author” section, enter your conflict of interest statement in the “Confidential to Editor” section, and submit your "Accept" recommendation.

Reviewer #2: All comments have been addressed

2. Is the manuscript technically sound, and do the data support the conclusions?

Reviewer #2: Yes

3. Has the statistical analysis been performed appropriately and rigorously? 

Reviewer #2: Yes

4. Have the authors made all data underlying the findings in their manuscript fully available?

Reviewer #2: Yes

5. Is the manuscript presented in an intelligible fashion and written in standard English?

Reviewer #2: Yes

6. Review Comments to the Author

Reviewer #2: Dear Authors

All comments seem to be addressed adequately. However, I would appreciate it if the authors brought out the issue of the use of Ceftriaxone/ cefotaxime use not being appropriate in their cohort that had a large number of Enterococci more explicitly. This would raise awareness in the reader when developing institutional guidelines.

7. PLOS authors have the option to publish the peer review history of their article (what does this mean?). If published, this will include your full peer review and any attached files.

Reviewer #2: **Yes: **Anusha Rohit

---

## [Editor Report · Acceptance letter]

14 Sep 2020

PONE-D-20-15175R1 

A Retrospective Analysis of cases of Spontaneous Bacterial Peritonitis in Cirrhosis patients 

Dear Dr. Angarone:

I'm pleased to inform you that your manuscript has been deemed suitable for publication in PLOS ONE. Congratulations! Your manuscript is now with our production department. 

Kind regards, 

on behalf of

Dr. Iddya Karunasagar 

Academic Editor

PLOS ONE